# Pressure induced superconductivity in MnSe

T. L. Hung [1,7], C. H. Huang [1,7], L. Z. Deng [2,7], M. N. Ou [1], Y. Y. Chen [1], M. K. Wu [1,3✉], S. Y. Huyan[2], C. W. Chu [2,4], P. J. Chen[5] & T. K. Lee [6]

The rich phenomena in the FeSe and related compounds have attracted great interests as it provides fertile material to gain further insight into the mechanism of high temperature superconductivity. A natural follow-up work was to look into the possibility of super-conductivity in MnSe. We demonstrated in this work that high pressure can effectively suppress the complex magnetic characters of MnSe, and induce superconductivity with $T_c$ ~ 5 K at pressure ~12 GPa confirmed by both magnetic and resistive measurements. The highest $T_c$ is ~ 9 K (magnetic result) at ~35 GPa. Our observations suggest the observed super-conductivity may closely relate to the pressure-induced structural change. However, the interface between the metallic and insulating boundaries may also play an important role to the pressure induced superconductivity in MnSe.

[1] Institute of Physics, Academia Sinica, Taipei, Taiwan. [2] Texas Center for Superconductivity and Department of Physics, University of Houston, Houston, TX, USA. [3] Department of Physics, National Tsing-Hua University, Hsinchu, Taiwan. [4] Lawrence Berkeley National Laboratory, Berkeley, CA, USA. [5] Institute of Atomic and Molecular Sciences, Academia Sinica, Taipei, Taiwan. [6] Department of Physics, National Sun-Yet-Sen University, Kaoshiung, Taiwan. [7] These authors contributed equally: T. L. Hung, C. H. Huang, L. Z. Deng. ✉email: mkwu@phys.sinica.edu.tw

The rich phenomena in Fe-based superconductors[1–5] have attracted great attention because the material has offered numerous insights into the mechanism of high-temperature superconductivity. The multiple-orbital nature of these materials, combined with spin and charge degrees of freedom, results in the observation of many intriguing phenomena, such as structural distortion, magnetic or orbital ordering[6], and electronic nematicity[7–11].

The parent compounds of FeAs-based materials exhibit structural transitions from a high-temperature tetragonal phase to a low-temperature orthorhombic phase, which accompanies with an antiferromagnetic (AFM) order[12,13]. Upon doping, both the orthorhombic structure and the AFM phase are suppressed and superconductivity is induced. On the other hand, FeSe undergoes a tetragonal-to-orthorhombic transition at ~90 K[2,14,15] without magnetic order at ambient pressure[15,16] and superconductivity below ~8 K[2,14] is crucially related to this orthorhombic distortion. The coexistence of nematic order with superconductivity without long-range magnetic order has led to arguments that the origin of the nematicity in FeSe is not magnetically but likely orbital-driven[17,18]. More recent studies show the application of pressure leads to the suppression of structural transition, the appearance of a magnetically ordered phase at ~1 GPa[16,19], and transition temperature ($T_c$) increases to a maximum of about 37 K[20–25] at ~6 GPa.

A natural follow-up work was to look into the substitution effects of Fe by other transition metals on superconductivity of FeSe. We reported that substitution of up to 6% Mn to Fe does not affect much the superconductivity in FeSe[26]. On the other hand, only 3% Cu substitution to Fe completely suppressed the superconductivity of FeSe. It is known that MnSe forms in a cubic structure at ambient condition and exhibits anomalous magnetic structure[27,28] so that no superconductivity could be detected. The NiAs-type FeSe favors to form hexagonal $\gamma$-Fe$_{1-x}$Se that exhibits both antiferromagnetism and ferrimagnetism depending on composition[29]. And superconductivity only exists when FeSe forms tetragonal structure[2]. Therefore, it will be valuable to investigate whether one could manage to form MnSe with crystal symmetry favorable for superconductivity.

MnP, which has an orthorhombic structure (with Pbmn symmetry), was found to be the first Mn-based superconductor with transition temperature ~1 K under 8 GPa[30]. It is noted that at ambient pressure MnSe exhibits very much the same magnetic behaviors[31] as those observed in MnP[30]. Therefore, it is of great interest to investigate whether superconductivity can also be induced in MnSe system. An idea to test such a possibility is to use the smaller ion sulfur to replace selenium to generate internal pressure. Thus, we have carried out the detailed structural study of Mn(Se-S) system[31].

Based on the refined lattice parameters of the Mn(Se-S) system[31], we estimated the equivalent compression pressure (E.C.P.) in MnSe by systematic sulfur substitution, using the third-order Birch–Murnaghan equation of state reported by Catherine McCammon[32]. The results suggest the E.C.P. of MnS (relative to MnSe) is ~13.2 GPa as shown in Supplementary Table 1. The estimated E.C.P. of MnS relative to MnSe is higher than that required to induce superconductivity in MnP compound. However, no superconductivity was observed in MnS as it maintains in cubic phase with an AF-like order at ~150 K and an anomalous ferromagnetic-like order at ~16 K. Nevertheless, the results demonstrated that partial substitution of Se by S could indeed effectively suppress the partial transformation of the cubic phase to hexagonal phase[31].

Wang et al. showed in their study of MnSe that the lattice collapsed under high pressure[33]. In their studies, the crystal structure of MnSe distorts to orthorhombic phase with space group Pnma under ~30 GPa[33]. And this orthorhombic phase is identical to the MnP superconducting phase. They also showed the compound is in low spin states under pressure based on X-ray emission spectroscopy and the transport measurement on MnSe indicated the sample becomes metallic at ~30 GPa. However, the temperature and pressure range in their study were rather limited. Therefore, it is desirable to carry out a more detailed investigation on MnSe over a wide temperature range under high pressure.

## Results

**Pressure-induced superconductivity.** Figure 1a, b shows the temperature dependence of resistivity for MnSe at different pressures. The result at ambient pressure is consistent with the previous report[31]. In the pressure range <16 GPa, the sample exhibits semiconducting behavior. However, the semiconducting gap value decreases with increasing pressure. An abrupt drop in resistivity at room temperature (RT) was observed at ~10 GPa, and a second resistivity drop appears at pressure ~16 GPa, at which the sample changes to metallic behavior. Meanwhile, a small drop in resistance is observed at ~4 K above 16 GPa. This low-temperature drop becomes more prominent as pressure increases, showing clear superconducting transitions above 20 GPa. It is noted that a third drop in resistivity at RT occurs above 20 GPa and the data above 30 GPa show much larger residual resistance ratio. Figure 1c presents the detailed resistive transition of the sample with pressure above 20 GPa, showing the superconducting transition with zero resistance above 2 K at pressures above 36 GPa. Figure 1d displays the resistive transition under different magnetic fields.

The magnetic field effect further confirms the superconducting transition nature. Figure 1e displays the magnetic field dependence of $T_c$ for RT at 36 GPa. The upper critical field observed is ~3463 Oe determined by using the Werthamer–Helfand–Honenberg formula, $H_{C2}(0) = -0.693(dH_{C2}/dT)T_c$, where $dH_{C2}/dT$ is ~806 Oe/K and $T_c$ is 6.2 K. The estimated superconducting coherence length is $\xi \sim 308$ Å following the Ginzburg–Landau formula, $\mu_0 H_{C2}(0) = \Phi_0/2\pi\xi^2$. Figure 1f reveals the pressure dependence of $T_c$ where $T_c$ is defined by the intersection temperature of two adjacent fitting lines of $d\rho/dT$ as displayed in Supplementary Fig. 2. The $T_c$ initially rises as the pressure increases. The maximum $T_c$ of 6.5 K appears at ~40 GPa. Then $T_c$ decreases with increasing pressure above 40 GPa.

In magnetic measurements, we first reproduced the observations reported by Huang et al.[31] on MnSe at ambient condition in the pressure cell. Two magnetic anomalies were clearly detected in MnSe as displayed in Supplementary Fig. 3. One broad anomaly occurs between 100 and 200 K with a signal at the order of $10^{-4}$ emu/Oe·g, which was suggested to be a coupling results of the magnetic locking effect of $\beta$-MnSe and thermal fluctuation on the short-range ferromagnetic sheets in $\alpha$-MnSe. The other anomaly around 266 K was attributed to a partial transformation of the cubic phase to hexagonal phase. Here we defined the peak position for the anomaly between 100 and 200 K as $T_N$ and the peak position of the anomaly above 250 K as $T_s$. Both $T_N$ and $T_s$ increased as pressure increased up to 1.2 GPa, with $dT_N/dP \sim 18.9$ K/GPa and $dT_s/dP \sim 34.3$ K/GPa, as shown in Supplementary Fig. 5, which is different from the doping effect by replacing Se with S[30] (Details of the measured results can be found in Supplementary Fig. 3 and Supplementary Fig. 4.). As pressure increases, the amplitude of first anomaly first increases and then decreases, and then increases again, while the second anomaly first increases and then decreases.

**Pressure-suppressed anti-ferromagnetism.** For the high-pressure magnetization measurements using diamond anvil cell (DAC), we first measured the sample with zero pressure, and two

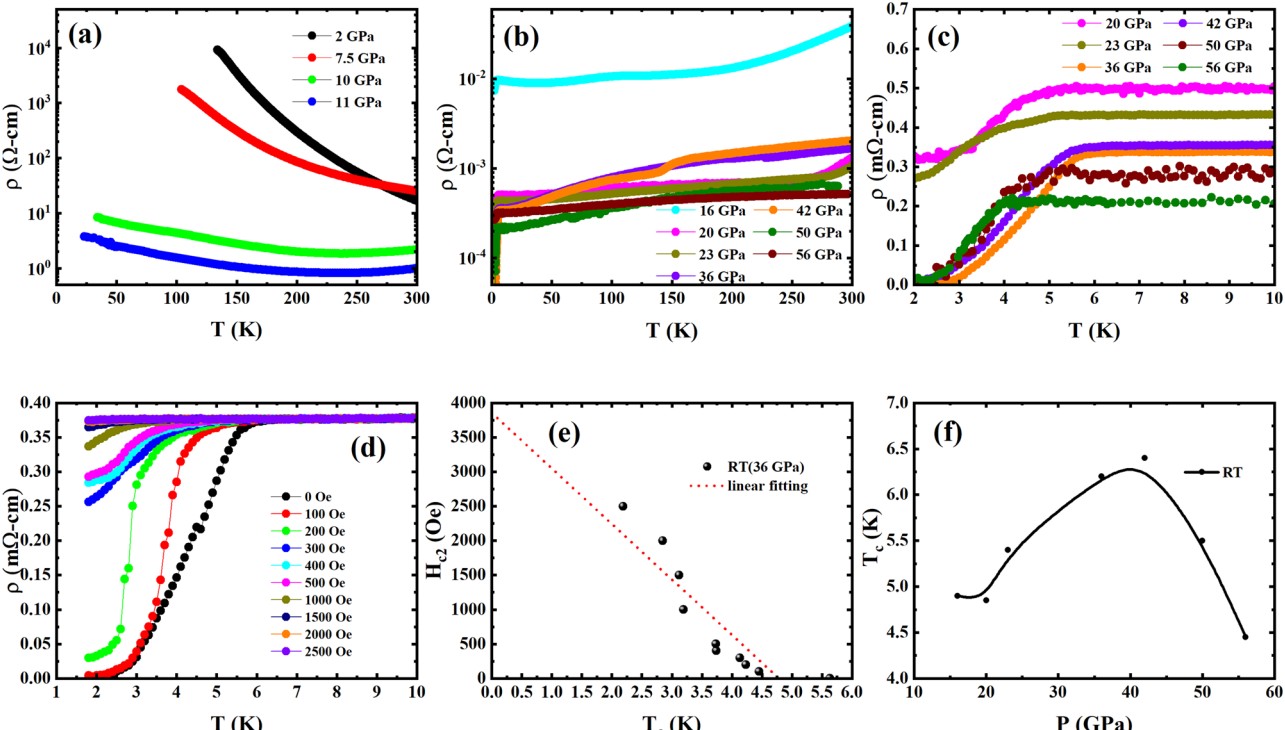

**Fig. 1 Temperature dependence of resistivity of MnSe at different pressures. a** below 11 GPa and **b** above 16 GPa where the resistance drop at low temperature appears. **c** shows the detailed resistive transition below 10 K, and **d** displays the resistive transition under different magnetic fields confirms the superconducting transition. **e** Field dependence of $T_c$ obtained from RT data at 36 GPa. **f** $T_c$ vs. $P$ from RT measurements, where $T_c$ is taken at the onset of resistive transition. The $T_c$ is obtained by the intersection of two adjacent lines fit of $d\rho/dT$ as described in Supplementary Fig. S2.

anomalies were both detectable (green line in Fig. 2a), though the signal size is small due to the small mass of the sample. At 2.68 GPa, the two anomalies were almost completely suppressed, while we observed an up-turn at lower temperature, which is similar to the results for $MnSe_{1−x}S_x$ at lower temperature[31], but no anomaly at around 150 K was observed at pressure below 11 GPa. As we continued to increase the pressure, a diamagnetic drop was observed at ≥11.75 GPa, as seen in Fig. 2b, which indicates possible pressure-induced superconductivity. The amplitude of the diamagnetic drop also increased as pressure increased. It is noted that a small hump was detected at pressure between 11.75 and 25.92 GPa at ~150 K, Fig. 2c, f, which is similar to the AFM transition reported in the $MnSe_{1−x}S_x$[31]. Above 25.92 GPa, the data suggest that the pressure suppresses the AFM transition, meanwhile the diamagnetic transition becomes more prominent, as shown in Fig. 2d.

The pressure dependence of Magnetization vs. Temperature (MT) measurements was used to determine the superconducting transition temperature ($T_c$) (shown in Supplementary Fig. 6-1, -2). $T_c$ vs. $P$ obtained from $M(T,P)$ data is summarized in Fig. 3a. A local minimum point appears around 26 GPa, which is consistent with a phase transition to be discussed later. The results exhibit certain inconsistency in the $T_c$ value, with the magnetic measurements showing relatively higher $T_c$ and richer behavior. It is noted that the superconducting transitions observed at various pressures are generally broad. However, the onset $T_c$, from either RT or MT measurements, as explained earlier, is well defined. Thus, the observation of higher $T_c$ onset by magnetization measurements is an experimental fact. The difference of the $T_c$ values derived from resistive and magnetic susceptibility measurements most likely was due to the pressure inhomogeneity with different DAC cells (from two different

laboratories) and pressure medium used. Additionally, observation of zero resistance depends on having a percolative path of superconductivity across the sample, whereas a diamagnetic response only needs a shell with thickness of order a penetration depth around isolated structural domains or grains. Figure 3b illustrates the magnetic field dependence of $T_c$ where $T_c$ is defined by $d\chi/dT$ vs. $T$ as shown in Supplementary Fig. 6-2. The magnetic field effect further confirms the superconducting transition nature in the MT data.

The effect of pressure homogeneity on superconducting is a complex issue. For example, Matsubayashi et al.[34] reported that superconductivity in $Bi_2Te_3$ was very sensitive to hydrostatic condition of the applied pressure and demonstrated that the superconducting phase could only survive under strong uniaxial stress. Another example is the pressure-induced superconductivity in $CaFe_2As_2$, which could only be observed above 0.5 GPa by measurements using organic medium[35]; on the contrary, no superconductivity was observed in a helium medium[36]. Miyoshi et al. reported in FeSe system that the increase in $T_c$ is suppressed under non-hydrostatic pressure[37]. And much earlier work by Klotz and Schilling found that $T_c$ of Bi2212 is suppressed faster under hydrostatic conditions[38]. We used hexagonal-BN powder as the pressure medium for $R(T)$ measurements. Inevitably we expect to generate uniaxial stress inside the cell. This characteristic was reflected in the broadening of the ruby R1 peak[39], as shown in Supplementary Fig. 7. The $M(T)$ measurements used the mixture of methanol and ethanol in a ratio of 4:1 as pressure medium, which is expected to exhibit better hydrostatic condition.

**Structural transformation at pressures.** To gain more insight into the origin for the observed pressure-induced superconductivity, we performed a series of X-ray diffraction (XRD) measurements on

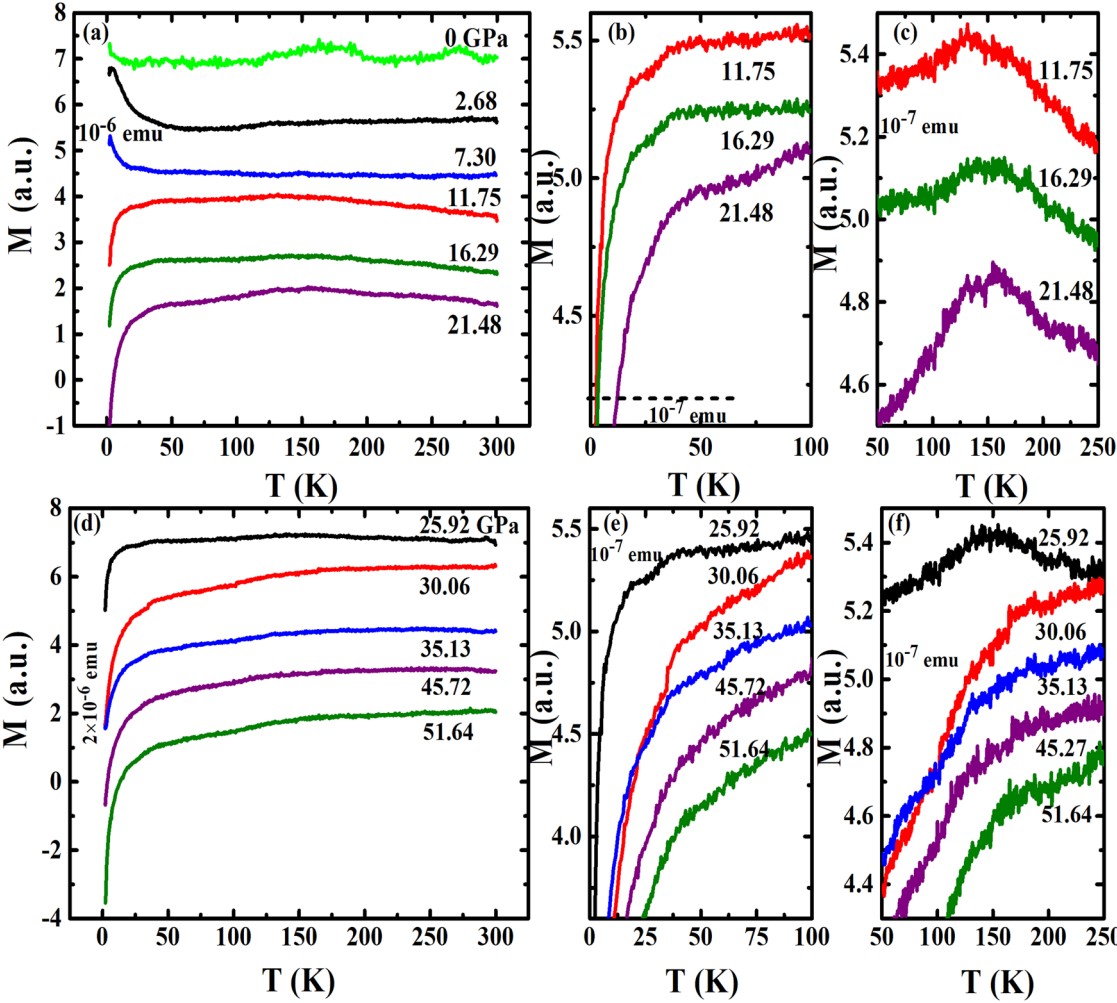

**Fig. 2 Temperature-dependent magnetic moments of MnSe at different pressures. a–c** $M$ vs. $T$ at different pressures up to 21.48 GPa. **d–f** $M$ vs. $T$ at different pressures between 25.92 and 51.64 GPa.

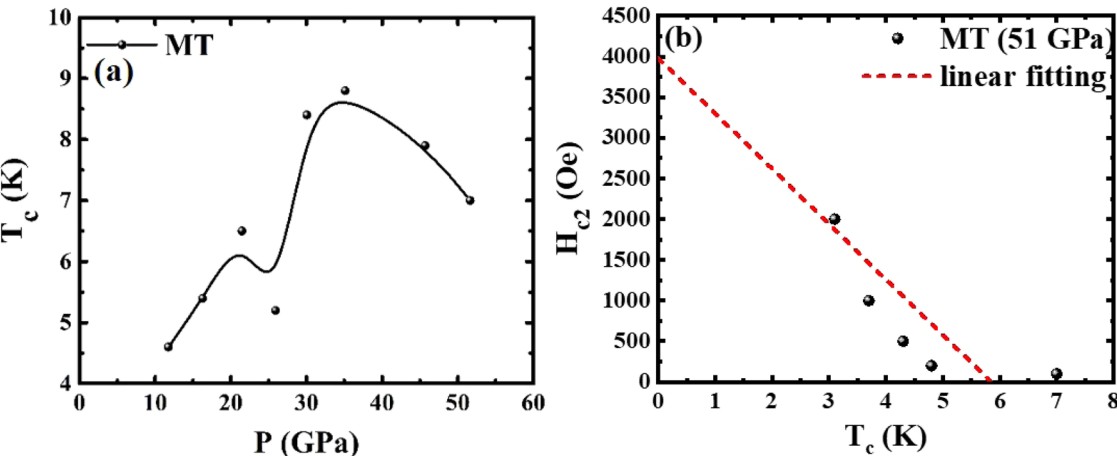

**Fig. 3 The $T_c$ evolution on pressure and the phase boundary in the $H$–$T$ plane. a** $T_c$ vs. $P$ from MT results. For MT measurement, $T_c$ is determined from by $dM/dT$ vs. $T$ as described in Supplementary Fig. 6-1. **b** The magnetic field dependence of $T_c$ for MT at 51 GPa.

MnSe under pressures. The results are presented in Fig. 4a. We used the helium gas as pressure transmitting medium (PTM) because of its excellent hydrostatic property up to at least 50 GPa[40]. It is noted that Wang et al. was using neon gas as PTM in their structural study on MnSe under pressure[33]. The crystal structure of MnSe is rock salt (cubic, Fm3m) with lattice constant 5.4697 Å at ambient condition. The results clearly show that MnSe undergoes two structural transformations at 12.2 and 30.5 GPa, respectively. The diffraction pattern exhibits the coexistence of cubic phase and hexagonal phase at 12.2 GPa, which is slightly higher than that reported by McCammon who showed the partial transformation at 9 GPa[32]. A surprising observation, which was not reported by

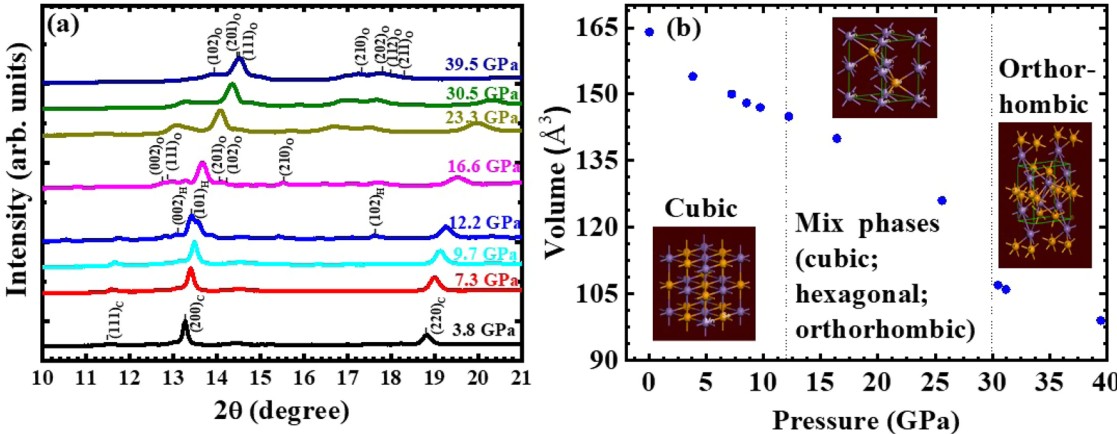

**Fig. 4 The XRD at pressures in MnSe. a** In situ synchrotron XRD patterns of MnSe during compression at room temperature; **b** the pressure dependence of volume for MnSe. The initial structure is cubic phase, and the intermediate structure retains cubic phase and exhibits the new hexagonal and orthorhombic phases, and the final structure is orthorhombic phase, which could be a new superconducting phase.

earlier studies, was that the sample was shown to exhibit another partial transition at ~16 GPa. The new phase present is orthorhombic so that within the pressure ranges from 16 to 30 GPa the sample is in a mixed state with the coexistence of cubic, hexagonal, and orthorhombic phases, as shown in Fig. 4. The X-ray patterns show another structural transformation at 30 GPa that MnSe completely transformed to orthorhombic phase (MnP-type, Pnma) with lattice $a = 5.7527$ Å, $b = 3.1045$ Å, and $c = 6.0434$ Å. These data are consistent with the results from theoretical calculations though our refinement gave slightly larger lattice parameters.

For a better comparison with the MT results, we have further performed the high-pressure XRD (HP-XRD) using M:E 4-1 as PTM. The observed results and analysis are displayed in Supplementary Fig. 8. According to Klotz et al., the M:E 4-1 liquid PTM at RT becomes solid at pressure of ~11–12 GPa[41]. However, the results of HP-XRD with M:E 4-1 as PTM clearly confirm the transformation of structure from cubic to hexagonal at ~12 GPa as shown in Supplementary Fig. 8-2. In addition, the orthorhombic phase also appears at about the same pressure (as shown in Supplementary Fig. 8-2) with the M:E 4-1 PTM instead of appearing at ~16 GPa in helium PTM. This observation indicates that the emergence of orthorhombic structure is closely associated with the appearance of superconductivity observed by MT measurements, which use the same M:E 4-1 PTM. Furthermore, the pressure dependence of volume for MnSe with liquid as PTM, as shown in Supplementary Fig. 8-3, indicates that the transformation of MnSe to single orthorhombic phase happens at lower pressure comparing with that in helium gas PTM.

To support the experimental observations, we made calculations to estimate the energetically favored phase under pressure. The results are shown in Supplementary Table 3. The results show that the cubic phase is most energetically favored at pressures <10 GPa, while the hexagonal (H) and orthorhombic (O) phases have about the same energy in the range of 10–30 GPa and it is only slightly lower than the cubic (C) phase. This is consistent with the phase diagram of Fig. 4b that C, H, and O phases are all mixed between 12 and 30 GPa. At around 40 GPa pressure, O phase has much lower energy than the other two phases.

We have further analyzed the HP-XRD data based on the Scherrer equation and Williamson–Hall method[42] to estimate the variation of mean grain size and micro-strain of MnSe above the pressure where it shows only single orthorhombic phase. Respectively, Supplementary Fig. 9-1, -2 represents the fitting

curve of each pattern obtained using helium gas as PTM, and the calculated results are listed in Supplementary Table 2. The mean size of the orthorhombic phase seems to remain about the same (though with small increase). And the calculated micro-strain generated under pressure at the single orthorhombic phase using gas PTM is in the order of 1% but increases with increasing pressure in the pressure range investigated.

The correlation between the superconducting $T_c$ with lattice strain of materials has been investigated either in the presence of defects[43] or under pressure[44]. The increase in lattice strain due to the presence of Mg vacancies was found to supress $T_c$ of $MgB_2$[43]. On the other hand, superconducting $T_c$ of rhenium under pressure was found to enhance along with increasing lattice strain. Detailed structural analysis of Rh case shows the grain size is reduced accompanying unit cell expansion suggesting that the enhancement in $T_c$ is due to the shear stress[44]. The rhenium result is in line with the observed $T_c$ increasing in MnSe at the pressure range where the material is orthorhombic. However, in contrast, the lattice of the MnSe shrinks with increasing pressure. Therefore, the observed pressure dependence of $T_c$ in MnSe in the range of 30–40 GPa cannot be simply associated with the pressure-induced orthorhombic phase. It is noted that similar pressure enhancing $T_c$ meanwhile suppressing the lattice were reported on FeSe superconductor[45,46].

**Local-density approximation (LDA) calculation for orthorhombic MnSe.** At ambient pressure, our calculations indicate that MnSe is cubic with AFM configuration consistent with that reported[31]. Besides, a structural phase transition from hexagonal to orthorhombic MnSe at pressure ~40 GPa is found, which agrees with the experimental observation. The crystal and band structures of the orthorhombic phase at 40 GPa are shown in Fig. 5. It is also noted that MnSe shows the low-spin state ($S = 0.5$) in the orthorhombic phase as reported[28].

## Discussion

The partial cubic-to-hexagonal transformation at 12 GPa could be similar to the low-temperature stress-induced transformation of 30% cubic-to-hexagonal structure at ambient condition, which has been extensively investigated and is understood to be the source for anomalous magnetic observed in MnSe at ambient condition[31]. The hexagonal phase remains to be insulating so that it could not be the source for the observed pressure-induced superconductivity. The observation of the orthorhombic phase

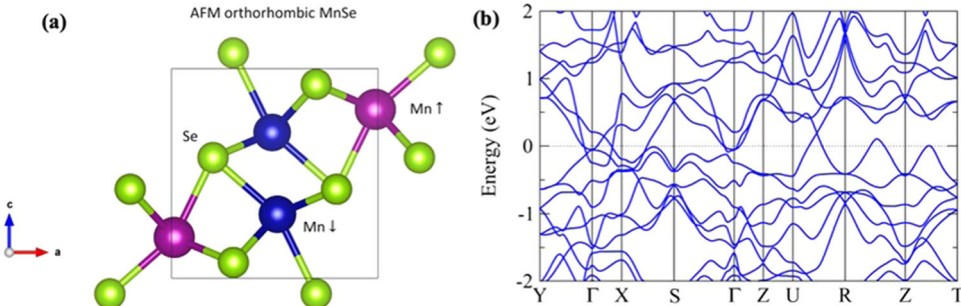

**Fig. 5 The band structure calculation for orthorhombic MnSe. a** The crystal structure of orthorhombic MnSe. Purple, blue, and green spheres represent Mn, Mn, and Se atoms, respectively. **b** The band structure of orthorhombic MnSe with $a = 5.280$ Å, $b = 2.999$ Å, and $c = 5.472$ Å.

appears at ~16 GPa, which coincides with the appearance of metallic behavior from the resistive measurements. It is also the pressure at which the onset of superconducting resistive transition is identified, though the magnetization measurements suggest the transition appears at lower pressure (~12 GPa).

Based on the results of HP-XRD studies, one would suggest that pressure-induced superconductivity is connected to the observed orthorhombic phase appeared at high pressure. If this were the case, one would expect the diamagnetic signal after 30 GPa would substantially increase as the material become orthorhombic single phase. However, the observed diamagnetic signals below and above the structural transition are comparable. Furthermore, one would also expect the $T_c$ values determined by MT and RT results after 30 GPa would be the same if superconductivity is associated with the orthorhombic phase. In fact, the observed results show that the difference in onset $T_c$ by two different methods is even larger above 30 GPa.

Normally, one would expect to observe the resistive onset $T_c$ to be either the same or higher than that obtained from magnetization measurements. The observation of a higher $T_c$ by magnetic measurements in pressurized MnSe is rather unusual. It has been reported in a $K$-doped FeSe superconductor ($K_2Fe_{4+x}Se_5$ system) that the magnetic transition temperature is higher than the resistive transition temperature[47,48]. For example, both the magnetization and resistive measurements show consistently an onset $T_c \sim 31$ K for the samples with $x = 0.2$ prepared with rapid quenching directly after annealing at 850 °C displaying high superconducting volume fraction. However, for the sample after post-annealed at 400 °C for 2 h, the $T_c$ determined by magnetization measurement remains with an onset at 31 K, but with much smaller volume fraction, the resistive transition was suppressed to a $T_c$ onset at 21 K. The longer the low-temperature annealing time, the smaller the superconducting volume fraction and lower resistive superconducting transition $T_c$. This result was due to the presence of a high volume of non-conducting phase in the post-annealed sample.

It is noted that diamagnetic susceptibility up to 45 K can only be observed in the alternating current susceptibility at high frequency in ultrathin FeSe films due to the possible interface-enhanced superconductivity[49]. The observation of relatively low upper critical field in MnSe might provide additional support to the picture of interfacial effect as the observed transition could be due to the Josephson junction coupling between grains[50].

In summary, we have undoubtedly demonstrated the pressure-induced superconductivity in MnSe. The anomalous magnetic behavior of MnSe at ambient condition was quickly suppressed by applying pressure. Superconductivity kicks in at ~12 GPa as shown by magnetic measurement (and at ~16 GPa by resistive measurement). The appearance of the superconducting transition $T_c$ coincides nicely with the appearance of orthorhombic phase. The transition temperature in MnSe under pressure is much

higher than that of the pressure-induced superconductivity in MnP[30] though they may exhibit the same crystalline phase under high pressures. A local minimum point appears around 26 GPa by magnetic measurements. Though it is very possible that the pressure-induced superconductivity is associated with the pressure-induced orthorhombic phase, however, our data suggest that the interfacial effect between the metallic and insulating boundaries may play an important role in the induced superconductivity.

## Methods

**Material syntheses.** Polycrystalline MnSe samples were prepared by solid-state reaction method using raw materials of Mn (99.95%, Alfa-Aesar) and Se (99.95%, Acros-Organic). The stoichiometric mixture of these elements was sealed in an evacuated quartz ampoule. The mixture was slowly heated to 750 °C, annealed for several hours, and then furnace-cooled to RT.

**Structural characterization.** Angle dispersion XRD (ADXRD) experiments were performed using a symmetric DAC with 300 μm culets. A rhenium gasket was pre-indented to a thickness of ~50 μm from an initial thickness of 250 μm. A 150-μm-diameter sample chamber was drilled in the center of the pre-indented gasket. Micro-meter ruby balls were placed inside the sample chamber as the pressure gauge[51]. Helium gas was used as a PTM using a gas loading system. ADXRD measurements were collected using the beamline BL01C2 at the National Synchrotron Radiation Research Center (NSRRC), Taiwan. The X-ray energy was 20 keV.

**Resistivity measurement.** High-pressure resistivity measurements used a DAC with 400 μm culets. A rhenium gasket was covered by cubic-BN powders for insulating the electrical leads. A MnSe of dimension ~70 μm × 70 μm × 15 μm is loaded in a sample chamber filled with hexagonal-BN as PTM. Gold foils were used as electrodes to connect the sample and gold wires. Details of the contact arrangement are shown in Supplementary Fig. 1. Pressure was determined by ruby florescent method[51]. Resistance measurements at low temperature was measured by Van der Pauw method in the $^4$He cryostat.

**Susceptibility measurement.** A mini-DAC fabricated from BeCu alloy, which was adapted into a Quantum Design Magnetic Property Measurement System (MPMS), was used for ultrasensitive magnetization measurements under high pressures[52]. A pair of 300-μm-diameter culet-sized diamond anvils was used. The gaskets were made from nonmagnetic Ni–Cr–Al alloy. Each gasket was pre-indented to ~20 μm in thickness, and an ~120-μm-diameter hole was drilled to serve as the sample chamber. The mixture of methanol and ethanol in a ratio of 4:1 (M:E 4-1) was used as the PTM. The applied pressure was measured by the fluorescence line of ruby powders. A piston-cylinder-type high-pressure cell, compatible with MPMS, was used when performing low-pressure measurements up to 1.3 GPa, where the pressure medium was Daphne-7373 oil and the pressure manometer was a lead piece.

**Theoretical calculation.** The first-principles calculations are performed using Quantum Espresso[53] with norm-conserving LDA pseudopotentials. The energy cut-off for the plane-wave expansion is 60 Ry. At low pressures, MnSe is semiconducting, so Coulomb $U = 5$ eV is included to deal with the correlation of the localized $3d$ orbitals. At higher pressures, both hexagonal and orthorhombic MnSe are metallic. The Coulomb $U$ is not included because it is unimportant in metallic systems where the orbitals are more delocalized.

## Data availability

All data supporting the findings of this work are available from the corresponding author on request.

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

## Acknowledgements

The authors at Academia Sinica thank the financial support from the Academia Sinica Thematic Research Grant No. AS-TP-106-M01 and the Ministry of Science and Technology of Taiwan Grant No. MOST108-2633-M-001-001 and No. MOST108-2112-M-001-042. The work performed at the Texas Center for Superconductivity at the University of Houston is supported by the U.S. Air Force Office of Scientific Research Grant FA9550-15-1-0236, the T. L. L. Temple Foundation, the John J. and Rebecca Moores Endowment, and the State of Texas through the Texas Center for Superconductivity at the University of Houston.

## Author contributions

T.L.H., C.H.H., L.Z.D., and M.K.W. conceived the project. C.H.H. synthesized the powder and crystal samples. T.L.H., M.N.O., and Y.Y.C. performed the high-pressure resistive measurement. T.L.H. and C.H.H. performed the in situ XRD, and C.H.H. performed quantitative analyses of the XRD results. L.Z.D., S.Y.H., and C.W.C. performed the high-pressure magnetic measurements. P.J.C. and T.K.L. performed the theoretical calculation. T.L.H., C.H.H., L.Z.D., and M.K.W. wrote and revised the paper. All authors participated in discussing the results and editing the manuscript.

## Competing interests

The authors declare no competing interests.

**Additional information**

