## [Peer Review File · Nature Communications]

REVIEWER COMMENTS

Reviewer #1 (Remarks to the Author):

Huang et. al report superconductivity in MnSe with T_c around 5-9K under high pressure from 16 to 56 GPa by transport and magnetization measurements. The authors also show that MnSe undergoes several structural transformations from a cubic phase under ambient pressure to coexistence of cubic, hexagonal and orthorhombic phases from 12 GPa to 30 GPa, and a single orthorhombic phase above 30 GPa. They suggest that the superconductivity is connected to the orthorhombic phase at high pressures.

The discovery of superconductivity in MnP has aroused considerable interest in exploring Mn-based superconductors. As far as I know, MnSe is the second Mn-based material that exhibits superconductivity and its superconducting transition temperature is higher than that of MnP. The findings in the manuscript are important and would stimulate searching rare Mn based superconducting compounds with higher T_c . I recommend publication of this paper in Nature Communications after considering my following point:

The authors estimated the upper critical field $H_{c2}(0)$ by simply extrapolating to zero temperature may be not accurate because normal $H_{c2}(T)$ exhibits negative curvature at low temperatures. It is better to estimate $H_{c2}(0)$ by Werthamer-Helfand-Hohenberg formula or other expressions. Besides, it seems superconductivity in this material is more easily destroyed in magnetic fields compared with its relatively higher T_c . As shown in fig 5b of the manuscript, a small field of 2500(Oe) can suppress T_c by over 3K. I suggest the authors could give some explanation why H_{c2} is small.

Reviewer #2 (Remarks to the Author):

Review on " Pressure induced superconductivity in MnSe" by Hung et al.

Hung et al. reported the pressure induced superconductivity in MnSe, the second Mn-based superconductor after MnP by means of electric and magnetic property measurements. They claimed the superconducting emerged after the structure phase transition from cubic to hexagonal to orthorhombic transition, and possibly the interface between the metallic and insulating boundaries. Overall, the authors provided solid evidence of experimental observation of superconductivity with transport measurements, but many aspects remain unclear. Before making a recommendation, I like to see a more detail discussion on several issues list below.

- 1) Resistivity measurements were conducted with hBN as pressure transmitting medium (PTM), and a BeCu cell; magnetic measurements below 1.3 GPa were conducted with Mini-DAC QD-MPMS, BeCu cell, 4:1 M/E PTM, and Daphne-7373 oil PTM above 16 GPa. The authors claimed that the superconductivity could largely depend on the interface among three phases (cubic, hexagonal and orthorhombic phase), so the hBN was chosen for generating high uni-axial stress. But both M:E 4-1 and Daphne 7373 are considered as reasonable good hydrostatic PTM in general. So the comparison of $R(T)$ and $M(T)$ using different PTM is not fair, and the conclusion needs to be adjusted.
- 2) Ref. 32 shows a High-Spin (HS) to Low-Spin (LS) transition at near 30 GPa accompanying with a cubic (C) to an intermediate unknown phase and then to orthorhombic (O) phase transition from 20 to 30 GPa, while the HP phase has LS state. Current paper further claimed the intermediate phase as hexagonal (H) structure. It is not clear which kind of PTM was used for the XRD measurement, and how one could unambiguously determine the HCP phase by only three weak peaks in Fig. 6. Can you provide theoretical evidence that it is energetic favor to take the path C-H-O phase? Did you try to

introduce the same uni-axial pressure environment for the XRD measurements?

3) In Fig. 2, it is suggested to have the same vertical scale, like T_n 160-210, T_s 260-310, so audiences can compare the slope immediately. What is the connection of the T_n , T_s measurements below 1.3 GPa with the rest of paper? T_n , T_s were suppressed by pressure within the cubic phase, but this suppression has no connection to the superconductivity since the cubic phase behaviors as semiconductor.

4) In Fig. 1, the resistivity scale from c to d is off by three orders of magnitude. Most likely the y-value should change unit to milli-ohm – cm. For 20 and 23 GPa $R(T)$ data, the resistivity did not fall to zero, from your XRD explanation, is there still in mixture of H-phase and O-phase and there is no through O-phase channel? Magnetic susceptibility measurement should be able to show the percentage of superconducting phase.

5) In Fig. 3C, there is no 25.92 GPa (you mentioned in text 11.75 GPa to 25.92 GPa) data, please make it up. How do you normalize the y-axis from Fig. 3(a) to 3(c)? Same question to Fig. 4(a) to 4(c).

6) Due to different PTM used in $R(T)$ and $M(T)$, the comparison in Fig. 5 (a) does not have much meaning. The comparison in Fig. 5(b) does not prove anything, it should include four sets of data, MT at 2 pressure points, and RT at similar 2 pressure points.

7) The interpretation of the interface strain mechanism of the superconductivity is very weak. The authors should provide clear evidence to show that. For example, run XRD with 7373 PTM and without any PTM, so the uni-axial stress effect could be enlarged. With the Scherrer's equation, one can fit the strain at each XRD pattern to make a claim.

8) The FeSe superconducting is in its low temperature orthorhombic phase, here the structure measurements were conducted at room temperature. It is worthy to check if there is any structural change during cooling down to superconducting phase. Otherwise, the hypothesis of interface between hexagonal phase and orthorhombic phase as superconductivity source is questionable.

We take this opportunity to thank all reviewers for carefully reading our manuscript and for the valuable comments/suggestions. We have considered all the reviewers' comments and suggestions to perform more experiments and revised our manuscript accordingly that have improved the scientific value of our work. The followings are the point-to-point responses to the reviewers' comments.

Reviewer #1 (Remarks to the Author):

Huang et. al report superconductivity in MnSe with T_c around 5-9K under high pressure from 16 to 56 GPa by transport and magnetization measurements. The authors also show that MnSe undergoes several structural transformations from a cubic phase under ambient pressure to coexistence of cubic, hexagonal and orthorhombic phases from 12 GPa to 30 GPa, and a single orthorhombic phase above 30 GPa. They suggest that the superconductivity is connected to the orthorhombic phase at high pressures.

The discovery of superconductivity in MnP has aroused considerable interest in exploring Mn-based superconductors. As far as I know, MnSe is the second Mn-based material that exhibits superconductivity and its superconducting transition temperature is higher than that of MnP. The findings in the manuscript are important and would stimulate searching rare Mn based superconducting compounds with higher T_c . I recommend publication of this paper in Nature Communications after considering my following point:

The authors estimated the upper critical field $H_{c2}(0)$ by simply extrapolating to zero temperature may be not accurate because normal $H_{c2}(T)$ exhibits negative curvature at low temperatures. It is better to estimate $H_{c2}(0)$ by Werthamer-Helfand-Honenberg formula or other expressions. Besides, it seems superconductivity in this material is more easily destroyed in magnetic fields compared with its relatively higher T_c . As shown in fig 5b of the manuscript, a small field of 2500(Oe) can suppress T_c by over 3K. I suggest the authors could give some explanation why H_{c2} is small.

Our response:

We would like to thank the reviewer for supporting to publish this work on Nat. Commun, and valuable comments.

Following the reviewer's suggestion, we have estimated $H_{c2}(0)$ using Werthamer-Helfand-Honenberg formula, $H_{c2}(0)=-0.693(dH_{c2}/dT)T_c$. The result is shown in Figure 1. A $H_{c2}(0)$ of $\sim 3,650$ Oe is then extracted. This value is lower than our earlier reported value. Possible reasons for low apparent H_{c2} are: 1) existence of weak links; 2) demagnetization enhancement. The effective magnetic field is approximately l/d times of the applied field, where l is the diameter of

the sample and d is the sample thickness, since our sample can be regarded as a thin sheet plate during the high-pressure measurements. In our case for magnetic measurements, $l \sim 0.1$ mm and $d < 0.01$ mm. The shape in resistance measurement is the same as a thin sheet plate, which the diameter is 0.1 mm and the thickness $d < 0.015$ mm. It's noted that H_{C2} in MnP system is also relatively small.

Figure 1. Field dependence of T_c obtained from RT at 36 GPa. The dash line is linear fit to obtain the critical field.

Reviewer #2 (Remarks to the Author):

Review on “ Pressure induced superconductivity in MnSe” by Hung et al.

Hung et al. reported the pressure induced superconductivity in MnSe, the second Mn-based superconductor after MnP by means of electric and magnetic property measurements. They claimed the superconducting emerged after the structure phase transition from cubic to hexagonal to orthorhombic transition, and possibly the interface between the metallic and insulating boundaries. Overall, the authors provided solid evidence of experimental observation of superconductivity with transport measurements, but many aspects remain unclear. Before making a recommendation, I like to see a more detail discussion on several issues list below.

1) Resistivity measurements were conducted with hBN as pressure transmitting medium (PTM), and a BeCu cell; magnetic measurements below 1.3 GPa were conducted with Mini-DAC QD-MPMS, BeCu cell, 4:1 M/E PTM, and Daphne-7373 oil PTM above 16 GPa. The authors claimed that the superconductivity could largely depend on the interface among three phases (cubic, hexagonal and orthorhombic phase), so the hBN was chosen for generating high uni-axial stress. But both M:E 4-1 and Daphne 7373 are considered as reasonable good

hydrostatic PTM in general. So the comparison of $R(T)$ and $M(T)$ using different PTM is not fair, and the conclusion needs to be adjusted.

Our responses:

The reviewer is correct that pressure medium plays an important role in high pressure experiments, as reported in many publications [e.g., 1, 2]. We did try to use M:E 4-1 as the pressure medium to get a better hydrostatic condition. Unfortunately, the M:E 4-1 solution is causing degradation of the grain boundary contact, the typical result is shown in Figure 2, that prevent us from obtaining reliable results, which is not an issue for magnetic studies under high pressure. However, we also successfully performed the X-ray diffraction under high pressures using M:E 4-1 as PTM. The detailed results will be presented in our reply to Question 2.

Figure 2. Temperature dependence of resistance using mixture of M:E 4-1 as the pressure medium at 24 and 32 GPa.

[1] S. Klotz and J.S. Schilling. (1993) *Physica C* 209, 499-506.

[2] Yu et. al, (2018) *Journal of alloys and Compounds* 767, 811-819

2) Ref. 32 shows a High-Spin (HS) to Low-Spin (LS) transition at near 30 GPa accompanying with a cubic (C) to an intermediate unknown phase and then to orthorhombic (O) phase transition from 20 to 30 GPa, while the HP phase has LS state. Current paper further claimed the intermediate phase as hexagonal (H) structure. It is not clear which kind of PTM was used for the XRD measurement, and how one could unambiguously determine the HCP phase by only three weak peaks in Fig. 6. Can you provide theoretical evidence that it is energetic favor to take the path C-H-O phase? Did you try to introduce the same uni-axial pressure environment for the XRD measurements?

Our response:

As mentioned in our manuscript, Cemič *et al.* reported the atomic structure of MnSe at 9 GPa [3, 4]. In their results, Cemič *et al.* suggested MnSe transforms from B1 (cubic) to B8 (hexagonal, P63/mmc) at 9 GPa. On the other hand, our group (see Huang *et al.* [5]) reported that at ~240K on cooling more than 20 percent of the cubic MnSe phase transforms to hexagonal phase (P63/mmc) without external pressure. This hexagonal phase is identical to that reported by Cemič *et al.* By comparing carefully the diffraction pattern to those obtain at low temperature under ambient pressure, we are very confident that hexagonal phase (P63/mmc) indeed appeared at 12.2 GPa. Therefore, we have no doubt that the intermediate phase in MnSe above 12.2 GPa consists of the mixture of cubic and hexagonal phase.

Regarding to the question of the PTM used in XRD, we did not use hBN as the PTM in our reported XRD measurements, we used helium gas as pressure medium instead. To double check the results, we made additional measurements recently using the M:E 4-1 as PTM. The representative diffraction patterns at 9.5, 11 and 16 GPa are shown in Figure 3(a), and the results of Rietveld refinement are displayed in Figure 3(b). The new results are consistent with the previous reported data except the orthorhombic phase seems to appear at a much lower pressure.

In our theoretical calculations as shown in the table below, the cubic phase is most energetically favored at pressures < 10 GPa, while the hexagonal (H) and orthorhombic (O) phases have about the same energy in the range of 10~30 GPa and it is only slightly lower than the cubic (C) phase. This is consistent with the phase diagram of Fig. 7 that C, H and O phases are all mixed between 12 ~30 GPa. Around P~ 40GPa, O phase has much lower energy than the other two phases.

Table I. Total energy/formula unit in meV. The lowest energy at each pressure is set to 0.

P (GPa)	cubic	hexagonal	orthorhombic
0	0	4.9	4.8
10	19.3	0	0.4
20	3.8	0.3	0
30	13.1	0.5	0
40	476.9	92.9	0

[3] McCammon, C. (1991). *Physics and chemistry of minerals*, 17(7), 636-641.

[4] Cemič, L., and Neuhaus, A. (1972). *High Temp.-High Press.*, 4, 97-99.

[5] C. H. Huang *et al.*(2019). *J. Magn. Magn. Mater.* 483, 205.

Figure 3(a) X-ray diffraction pattern of MnSe using methanol and ethanol in a ratio of 4:1 as the pressure medium at 9.5, 11, and 16 GPa.

Figure 3(b) Typical Rietveld refinement of MnSe at 11 and 16 GPa.

The results show that the lattice of MnSe below 9.7 GPa remains to be cubic, and partial cubic phase is transformed to hexagonal phase at 12 GPa. However, in liquid PTM, in addition to the coexistence of cubic and hexagonal phases, an orthorhombic phase appears at 11 GPa. When the pressure increases to 16 GPa, the orthorhombic phase becomes dominant in MnSe. This new results suggest that the onset of orthorhombic structural transformation appears at ~11 GPa, which correlates well with the onset of superconductivity determined magnetically, which used the same pressure medium.

3) In Fig. 2, it is suggested to have the same vertical scale, like T_n 160-210, T_s 260-310, so audiences can compare the slope immediately. What is the connection of the T_n , T_s measurements below 1.3 GPa with the rest of paper? T_n , T_s were suppressed by pressure within the cubic phase, but this suppression has no connection to the superconductivity since the cubic phase behaviors as semiconductor.

Our response:

Thank the reviewer for the valuable comments. Following this suggestion, we updated Figure

2 accordingly. The reviewer is correct that there is no direct connection among T_n , T_s and superconductivity. However, the low-pressure results provide a nice comparison with the chemical doping effect on this compound. Nevertheless, we moved this section to Supporting Information.

4) In Fig. 1, the resistivity scale from c to d is off by three orders of magnitude. Most likely the y-value should change unit to milli-ohm – cm. For 20 and 23 GPa R(T) data, the resistivity did not fall to zero, from your XRD explanation, is there still in mixture of H-phase and O-phase and there is no through O-phase channel? Magnetic susceptibility measurement should be able to show the percentage of superconducting phase.

Our response:

We agree with the reviewer's suggestion and have modified the resistivity unit to "mΩ-cm" in both Fig. 1(c) and Fig. 1(d) of the revised manuscript. In Fig.1(d), our RT results undoubtedly reveal the resistivity fall to zero at 36 GPa, at which the sample exhibit only orthorhombic phase. The non-zero resistivity observed between 20-23 GPa, as the reviewer pointed out, is most likely due to the presence of H-phase and O-phase mixture. Unfortunately, due to the background signal and demagnetization enhancement effect, it would be difficult for us to estimate the accurate superconducting volume fraction based on our current magnetic measurement results.

5) In Fig. 3C, there is no 25.92 GPa (you mentioned in text 11.75 GPa to 25.92 GPa) data, please make it up. How do you normalize the y-axis from Fig. 3(a) to 3(c)? Same question to Fig. 4(a) to 4(c).

Our Response:

Thank the reviewer for carefully reading our manuscript. Please find 25.92 GPa in Fig. 2 (d, e, f) in the revised manuscript. We did not normalize the y-axis. The curves were shifted vertically for comparison in both Figure 2.

6) Due to different PTM used in R(T) and M(T), the comparison in Fig. 5 (a) does not have much meaning. The comparison in Fig. 5(b) does not prove anything, it should include four sets of data, MT at 2 pressure points, and RT at similar 2 pressure points.

Our response:

We have revised the Figures so that the RT and MT results are presented separately as suggested by the reviewer. Please refer to Figure (1) for RT data, and Figure (2) for MT results

in the revised manuscript.

7) The interpretation of the interface strain mechanism of the superconductivity is very weak. The authors should provide clear evidence to show that. For example, run XRD with 7373 PTM and without any PTM, so the uni-axial stress effect could be enlarged. With the Scherr's equation, one can fit the strain at each XRD pattern to make a claim.

Our response:

Thank the reviewer for the constructive suggestion. We have thus investigated in more details the XRD data in pressure above 30 GPa where the sample exhibits only orthorhombic phase. We chose three diffraction peaks of the orthorhombic phase in with indices (102),(201) and (111), and we utilized 3 pseudo-voigt functions to fit the broaden peak as shown in Fig. 4 (c) to (e). The fitting results on the (111) peak are listed in Table (a), including FWHM (deg.), FWHM(rad.), and peak center (deg.). And then we used Scherrer equation to estimate the variation of MnSe mean grain size under pressure, the results are also listed in Table 2 (a). The fitting results show that the mean grain size of the orthorhombic phase decreases with increasing pressures.

Figure 4 (a)(b)(c) Rietveld refinement of MnSe at 30.5, 31.2, and 39.5 GPa with helium as the pressure medium.

Table 2 (a). List of FWHM (deg.), FWHM(rad.), Peak center (deg.), and Mean grain size.

	FWHM (deg.)	FWHM (rad.)	Peak center (deg.)	Mean Grain size (Å)
30.5 GPa	0.2745	0.00478	14.35	117.64
31.2 GPa	0.2743	0.00479	14.42	117.40
39.5 GPa	0.3773	0.00658	14.50	85.47

We have further carried out the same analysis on the new XRD results using M:E 4-1 as PTM case. The new XRD clearly identify the phase changes under pressure. However, due to relatively poor peak resolution of the beamline used in this experiment, the observed peaks are much broader comparing with the earlier results. Therefore, we use only single pseudo-voigt function to fit this broaden peak. The fitting curves and results are shown in Fig. 5(1) to (4) and Table 2(b), respectively. Consistent with the earlier data, the new results also show the reduction of mean grain size with increasing pressure.

Based on these fitting results, the grain size of the single orthorhombic phase at pressures above 30 GPa becomes smaller with increasing pressure. If we consider superconductivity solely from the orthorhombic phase, these results seem not consistent with the observed pressure dependence of T_c , as shown in Figs. 1(f) and 3(a) in the revised manuscript, which shows T_c increases in the range of 30 to 40 GPa.

Figure 5. Single peak fitting of MnSe with M:E 4-1 as the pressure medium at (1) 16, (2) 21, (3) 28, and (4) 31 GPa.

Table 2 (b). List of FWHM (deg.), FWHM(rad.), peak center (deg.), and mean size from peak fitting results of orthorhombic phase with M:E 4-1 as the pressure medium.

	FWHM (deg.)	FWHM (rad.)	Peak center (deg.)	Mean Grain size (Å)
16 GPa	0.4244	0.00741	12.82	75.77
21 GPa	0.4599	0.00803	12.95	69.93
28 GPa	0.4809	0.00839	13.20	66.94
31 GPa	0.4695	0.00819	13.26	68.58

8) The FeSe superconducting is in its low temperature orthorhombic phase, here the structure measurements were conducted at room temperature. It is worthy to check if there is any structural change during cooling down to superconducting phase. Otherwise, the hypothesis of interface between hexagonal phase and orthorhombic phase as superconductivity source is questionable.

Our response:

Generally, the room-temperature X-ray can give useful information on the phase transitions under pressure, though the critical pressures may be different at lower temperature. We agree with the referee that it is worthy to carry out X-ray measurement at low T, and we would like to leave this part for future work.

REVIEWER COMMENTS

Reviewer #1 (Remarks to the Author):

The authors of this manuscript have properly answered the questions raised by both referees, and revised the manuscript accordingly. Now, I recommend to publish this work in Nature Communications.

Reviewer #2 (Remarks to the Author):

The authors have answered most questions I had during the first round review process. But for the interpretation of the interface strain mechanism of the superconductivity, it is still not quite conveniencing. Fitting the scherrer equation, one can get both grain size and the averaged strain. The author paid the effort on the peak width fitting and grain size determination, but the key parameter on the strain level is missing, which should be given along with the grain size. Please finish this part, and show the clear correlation with the superconductivity as the authors claimed.

We want to thank the reviewers for accepting our responses to their previous comments and recommend publication of our manuscript in *Nature Communication* with additional analysis of the data. We have thus followed the suggestion from reviewer #2 to estimate the average strain based on the X-ray data. We have included these additional results to the revised manuscript and the supplementary information. The followings are the point-to-point responses to the reviewers' comments.

Reviewer #1 (Remarks to the Author):

The authors of this manuscript have properly answered the questions raised by both referees, and revised the manuscript accordingly. Now, I recommend to publish this work

Reviewer #1 (Remarks to the Author):

Our response:

All authors thank very much the reviewer for recommending to publish our work in *Nature Communication*.

Reviewer #2 (Remarks to the Author):

The authors have answered most questions I had during the first round review process. But for the interpretation of the interface strain mechanism of the superconductivity, it is still not quite convincing. Fitting the Scherrer equation, one can get both grain size and the averaged strain. The author paid the effort on the peak width fitting and grain size determination, but the key parameter on the strain level is missing, which should be given along with the grain size. Please finish this part, and show the clear correlation with the superconductivity as the authors claimed.

Our Response:

All authors really appreciate the reviewer to suggest including the strain generated in the sample based on the X-ray diffraction patterns so that the data analyses are more complete. The followings are the results of our additional analyses:

We used the Scherrer equation and Williamson-Hall plot method ($\tau = \frac{K\lambda}{\beta \cos(\theta)}$, $\varepsilon = \frac{\beta}{4 \tan(\theta)}$, τ : mean size; ε : micro strain (unit less); K : sharp factor, 0.9; λ : wave length; β : FWHM (rad.); θ : diffraction center.) to estimate the variation of mean grain size and micro-strain of MnSe under pressure above that MnSe shows single orthorhombic phase. Respectively, **figure S9-(1) to figure S9-(3)** are the fitting curve of each pattern obtained using helium gas as PTM, and the calculated results are listed in **Table S2**; **figure S10-(1) to figure S10-(4)** and **Table S3** are the results obtained using the mixture of methanol and ethanol as PTM. Due to the difference in XRD resolution for different PTM cases, we respectively used three (for PTM Helium case) or one (for methanol and ethanol case) pseudo-voigt function to fit the diffraction peaks. The fitting results in both PTM cases show consistently that the mean size of the orthorhombic phase decreases with increasing pressures. On the other hand, the calculated micro-strain generated under pressure at the

single orthorhombic phase consistently increases with increasing pressure in gas PTM, whereas it remains about the same in liquid PTM.

Figure S9. Multi-peak fitting of MnSe with helium as the pressure medium at (1) 30.5, (2) 31.2, and (3) 39.5 GPa.

Table S2. FWHM (deg.), FWHM(rad.), peak center (deg.), volume (\AA^3), mean grain size (\AA) and micro strain from (111) peak obtained from analysis of orthorhombic phase with helium gas as the pressure medium. The corresponding T_c values are also listed in the same table. (*: The T_c value is extracted from the Fig.1-f)

	FWHM (deg.)	FWHM (rad.)	Peak center (deg.)	Volume (\AA^3)	Mean size (\AA)	Micro strain (10^{-3})	T_c (K)*
30.5 GPa	0.2745	0.00478	14.35	107.91	117.64	9.51	5.85
31.2 GPa	0.2743	0.00479	14.42	106.07	117.40	9.46	5.89
39.5 GPa	0.3773	0.00658	14.50	99.18	85.47	12.94	6.25

Figure S10. Single peak fitting of MnSe with M:E 4-1 as the pressure medium at (1) 16, (2) 21, (3) 28, and (4) 31 GPa.

Table S3. FWHM (deg.), FWHM(rad.), peak center (deg.), volume (\AA^3), mean grain size (\AA) and micro strain from (111) peak obtained from analysis of orthorhombic phase with helium gas as the pressure medium. The corresponding T_c values are also listed in the same table. (**: T_c values are extracted from the Fig.3-(a))

	FWHM (deg.)	FWHM (rad.)	Peak center (deg.)	Volume (\AA^3)	Mean size (\AA)	Micro strain (10^{-3})	T_c (K)**
16 GPa	0.4244	0.00741	12.82	120.20	75.77	16.48	5.4
21 GPa	0.4599	0.00803	12.95	120.45	69.93	17.68	6.5
28 GPa	0.4809	0.00839	13.20	118.81	66.94	18.14	6.8
31 GPa	0.4695	0.00819	13.26	117.93	68.58	17.62	8.4

REVIEWER COMMENTS

Reviewer #2 (Remarks to the Author):

Authors have paid much effort to work out on the micro-strain level from the XRD profile. Based on their report, I would think the micro-strain they provided are much over-estimated. As I raised the question in the last round review, the peak broadening in XRD is contributed by two effects: nano-scale grain size, and micro-strain from the powder sample. The Scherrer equation is usually used to estimate both effect as follows:

$$(\text{FWHM} * \cos(\theta))^2 = (\text{wavelength}/d)^2 + (\text{strain} * \sin(\theta))^2$$

Here the theta is the diffraction angle, which is half of the center angle of diffraction peak in the I vs. 2theta profile. d is the average grain size.

Current authors calculated the grain size and micro-strain individually by ignoring the second term. Since the particle size is approaching a few nanometers, the first term at right side of above equation may dominate the peak broadening effect, which leaves much small room from the strain contribution. So I would really like to see the realistic strain level (by the way, a 2% strain looks too high for me) in the materials as the author claimed the interfacial strain may play great role in superconductivity.

We want to thank the reviewer's insight and suggestion. We have thus followed the suggestion to estimate again the average strain based on the X-ray data. We have included these additional results to the revised manuscript and the supplementary information. The following is the response to the reviewer's comments.

Reviewer #2 (Remarks to the Author):

Authors have paid much effort to work out on the micro-strain level from the XRD profile. Based on their report, I would think the micro-strain they provided are much over-estimated. As I raised the question in the last round review, the peak broadening in XRD is contributed by two effects: nano-scale grain size, and micro-strain from the powder sample. The Scherrer equation is usually used to estimate both effect as follows:

$$(\text{FWHM} * \cos(\theta))^2 = (\text{wavelength}/d)^2 + (\text{strain} * \sin(\theta))^2$$

Here the theta is the diffraction angle, which is half of the center angle of diffraction peak in the I vs. 2theta profile. d is the average grain size.

Current authors calculated the grain size and micro-strain individually by ignoring the second term. Since the particle size is approaching a few nanometers, the first term at right side of above equation may dominate the peak broadening effect, which leaves much small room from the strain contribution. So I would really like to see the realistic strain level (by the way, a 2% strain looks too high for me) in the materials as the author claimed the interfacial strain may play great role in superconductivity.

Our Response:

All authors appreciate the reviewer's insight and suggestion. We have thus followed to carry out more detailed analyses on our X-ray data. However, due to the relatively poor resolution of the XRD experiment, we could not physically meaningful results in the measurements using liquid as pressure transmission medium (PTM) (the experiments were performed in a different synchrotron beamline). Therefore, we present only the results obtained from the measurements using gas as PTM, where single orthorhombic phase exists above 30 GPa. The following table shows the calculated results (we did not include the data at 30.5 GPa, which is almost the same as that under 31.2 GPa).

	Peak	FWMH (deg.)	FWMH (rad.)	Peak center (deg.)	d (nm)	ε (%)	T _c [*] (K)
31.2 GPa	(111)	0.285(4)	0.00497	14.418(1)	12.(7)	0.6(5)	5.85
	(102)	0.284(9)	0.00495	13.310(9)			
39.5 GPa	(111)	0.283(2)	0.00494	14.511(4)	14.(2)	1.5(8)	6.25
	(102)	0.279(9)	0.00487	13.337(9)			

Indeed, as expected by the reviewer, we did over-estimated the micro-strain in our previous response. The current results show that the mean size of the orthorhombic phase seems to remain about the same (though with small increase). And the calculated micro-strain generated under pressure at the single orthorhombic phase using gas PTM is in the order of 1%, but consistently (with our previous estimate) increases in the pressure range investigated.